# Are Experts Well-Calibrated? An Equivalence-Based Hypothesis Test

**DOI:** 10.3390/e24060757

**Published:** 2022-05-27

**Authors:** Gayan Dharmarathne, Anca M. Hanea, Andrew Robinson

**Affiliations:** 1Department of Statistics, University of Colombo, Colombo 00700, Sri Lanka; 2Centre of Excellence for Biosecurity Risk Analysis, School of BioSciences, The University of Melbourne, Parkville, VIC 3010, Australia; anca.hanea@unimelb.edu.au (A.M.H.); apro@unimelb.edu.au (A.R.)

**Keywords:** credible intervals, experts’ hit rates, experts’ calibration, equivalence test

## Abstract

Estimates based on expert judgements of quantities of interest are commonly used to supplement or replace measurements when the latter are too expensive or impossible to obtain. Such estimates are commonly accompanied by information about the uncertainty of the estimate, such as a credible interval. To be considered *well-calibrated*, an expert’s credible intervals should cover the true (but unknown) values a certain percentage of time, equal to the percentage specified by the expert. To assess expert calibration, so-called *calibration questions* may be asked in an expert elicitation exercise; these are questions with known answers used to assess and compare experts’ performance. An approach that is commonly applied to assess experts’ performance by using these questions is to directly compare the stated percentage cover with the actual coverage. We show that this approach has statistical drawbacks when considered in a rigorous hypothesis testing framework. We generalize the test to an equivalence testing framework and discuss the properties of this new proposal. We show that comparisons made on even a modest number of calibration questions have poor power, which suggests that the formal testing of the calibration of experts in an experimental setting may be prohibitively expensive. We contextualise the theoretical findings with a couple of applications and discuss the implications of our findings.

## 1. Introduction

Expert elicitation refers to employing formal procedures for obtaining and combining expert judgments, when existing data and models cannot provide required information for decision making in practice [1]. Subjective estimates from experts can play an important role in decision making within emerging contexts for which no data are available. The existing literature of applications across widespread areas speaks to the importance of expert elicitation in practice (see, e.g., [2]).

Expert judgements of unknown quantities of interest are often elicited in the form of best estimates accompanied by credible intervals, which capture expert uncertainty around the point estimates (e.g., [2,3,4]). Credible intervals can be elicited using a 3-step or a 4-step question format depending on the context, as follows.

A 3-step format for eliciting quantities (a 3-step format for eliciting probabilities rather than quantities asks for best estimates and upper and lower bounds without the percentile operationalization. However, this situation is outside the scope of our research) refers to the elicitation of best estimates together with an upper and lower bound, which are often associated with an upper and lower percentile of the expert’s subjective distribution that represents the uncertainty around the best estimate, often taken as a median (e.g., [5,6]). The elicited intervals are often taken as central credible intervals.

A 4-step format elicits the previously mentioned three estimates together with a fourth one that corresponds to the expert’s confidence that the true value of the yet unknown quantity falls between the upper and the lower elicited bounds (e.g., [4,7]). This confidence is then used to define the credible interval. If for example, an expert is 90% confident that the true value falls between the bounds, then their upper and lower bound form a 90% credible interval. It is important to note that on grounds of psychological theory one can expect the four-step procedure to yield better calibration than the three-step [8]. To aid further probabilistic modelling, it is often necessary to formalize these credible intervals as central and hence fix the upper and lower bounds to represent given percentiles. However, this need not be the case when coverage probabilities alone are of interest.

For the present analysis we assume that experts provide credible intervals for multiple quantities together with a confidence level (constant for all quantities) that the true values fall within the provided intervals. Since each question is answered with an interval, throughout the paper we refer to expert answers as *intervals*.

Whenever possible, expert performance is assessed based on a number of performance characteristics, including their calibration. For this reason, it is advisable to include *calibration* questions among target questions in expert elicitation exercises (e.g., [1]); these are questions for which the answers are, or will become known in the near future. Such questions are used to assess the calibration of the experts using various ways of calculating calibration.

Approaches to formally evaluate calibration, or formally test whether an expert is well calibrated vary. Cooke’s classical model [3] is often used for assessing expert calibration when eliciting percentiles of probability distributions. However, this approach does not align with our context, in which the bounds of the elicited intervals are not necessarily fixed (to pre-specified percentiles).

In its simplest form, calibration is evaluated based on whether or not the experts’ stated confidence level, which corresponds to the declared coverage for their credible intervals, aligns with the actual coverage (when realized/true values become available). This is a commonly used approach [4,7,9,10,11] that fits our context. It is often referred to as a *direct comparison* between experts’ observed *hit rates* and the specified confidence. The observed hit rates are the observed proportions of elicited intervals that contain the realized/true values of a given set of quantities. We acknowledge that the direct comparison is a just a descriptive comparison of observed hit rates and the specified confidence. However, it is often applied in practice. Therefore, it is important and instructive to assess its properties from a statistical point of view. Here, we argue that this direct comparison has poor statistical characteristics that can be addressed by formalizing and re-framing it as an equivalence-style hypothesis test.

Here, we evaluate the properties of the formalized direct comparison test with those of the corresponding formal equivalence test of a single binomial proportion using a simulation experiment. We undertake a power analysis that shows that the number of observations required to provide a reasonable probability of detecting a well-calibrated expert (using these tests) is prohibitively high when compared to the number of observations affordable in a real-life expert elicitation exercise. We then compare and discuss these proposals when applied to two real-life expert elicited data sets.

The remainder of this paper is organized as follows. Section 2 ‘Materials and Methods’ includes sub-sections that describe the properties of the direct comparison and equivalence-based hypothesis tests, simulation of data, and the real-life expert-elicited data sets used in the analysis. Analyses on the simulated data and real-life data will be presented under the ’Results’ Section 3. Finally, the ‘Discussion’ Section 4 summarizes the conclusions, limitations, and recommendations for future studies in brief.

## 2. Materials and Methods

### 2.1. Direct Comparison as a Hypothesis Test

When evaluating calibration using the direct comparison, the experts’ hit rates are considered as fixed or deterministic quantities, disregarding their random variation due to the elicitation context. However, this random variation will naturally lead to variation in experts’ calibration. Acknowledging this latter variation formally may be tackled by employing a formal statistical testing procedure when estimating hit rates. [12] provide evidence that model based techniques (as opposed to raw relative frequencies/counts calculations) show great promise in the context of judgment and decision making. Furthermore, measuring experts’ calibration needs a better estimation procedure that can produce confidence (or credible) intervals around the estimated hit rates. Confidence intervals and hypothesis testing are interconnected as both are inferential techniques which use the estimate of an unknown population parameter to find a range of possible values that is likely to capture the unknown parameter and test the strength and validity of a hypothesis. Hypothesis testing can also be performed directly by using confidence intervals. Hence, in this paper, we develop an equivalence-based hypothesis testing approach to test experts’ calibration (see Section 2.2).

We first embed the direct comparison in a statistical testing framework. The direct comparison involves comparing the expert’s realized hit rate against their claimed coverage for a small number of calibration questions (denoted by *n*). Most often in applications, *n* is smaller than 20. Denote the claimed coverage by $P_{r}$, where $0<P_{r}<1$, and define *x* as the number of calibration questions for which the expert’s credible interval covers the true value. Then the direct comparison can be interpreted as a test: the expert is well-calibrated if $x=P_{r}\cdot n$ and is not well-calibrated otherwise.

Some disadvantages of this approach are immediately apparent, for example if *n* is not some convenient number such that $P_{r}\cdot n$ is an integer, then it is impossible for the expert to be considered perfectly well-calibrated. Furthermore, the test suffers from a peculiar statistical characteristic, namely it is easier for an expert to be declared well-calibrated when *n* is small than when *n* is large, so *the power of the test decreases with its sample size*. This unpleasant characteristic is evident in a subsequent analysis (see Section 3.1.1) after re-framing the direct comparison to an equivalence-style hypothesis test of a single binomial proportion in the following Section 2.2.

### 2.2. Equivalence-Based Hypothesis Test

The main scope of this research is to identify an appropriate statistical procedure to evaluate expert calibration that reduces the error of identifying a poorly-calibrated expert as well-calibrated. This error can be considered as more serious than the error of not identifying a well-calibrated expert as well-calibrated.

Consider applying the commonly used equality test of a binomial proportion to test expert calibration. The null and alternative hypotheses of the equality test relevant to this context can be stated as follows.
$$\begin{array}{cc} H_{0}:\; & P=P_{r}\\ H_{1}:\; & P\neq P_{r}; \end{array}$$
where *P* and $P_{r}$ denote the true unknown coverage and claimed coverage, respectively.

This is the usual hypothesis testing paradigm that assumes the expert is well calibrated as the null hypothesis against the alternative hypothesis that the expert is not well calibrated. Non-rejection of the null hypothesis does not statistically imply that the null hypothesis is true. Hence, this form of hypothesis testing does not match with our objective of reducing the error of identifying a not well-calibrated expert as well calibrated. For this reason, we consider the equivalence test of a single binomial proportion instead; rather than using the usual hypothesis testing paradigm to assume that the expert is well-calibrated as the null hypothesis above, we consider the null hypothesis to be “the expert is not well-calibrated” against the alternative “the expert is well-calibrated”.

We now review the exact version of the equivalence test of a single binomial proportion presented in (Chapter 4 in [13]) in place of the large-sample approximation test. Doing so allows to keep the flexibility for considering both small and large number of elicited intervals in the analysis. For convenience, hereafter we refer to the equivalence test of a single binomial proportion just as the *equivalence test*, and the direct comparison of hit rates as the *direct test*. Further details on practically applying the equivalence tests can be found in [14].

Suppose that an unknown population proportion of success *P* is required to be statistically tested for equivalence with a reference value $P_{r}$. Equivalence can be claimed if *P* remains between $P_{1}=P_{r}-\epsilon_{1}$ and $P_{2}=P_{r}+\epsilon_{2}$, where an acceptable margin of deviation around the reference value is allowed in the test. $P_{1}$ and $P_{2}$ can be made symmetric around $P_{r}$ by taking $\epsilon_{1}=\epsilon_{2}=\epsilon$, unless specific, different values are preferred for $\epsilon_{1}$ and $\epsilon_{2}$, which will be context dependent. In practical terms, this acceptable margin ϵ should be the maximum difference that one is willing to accept to declare equivalence when the data provide enough evidence to conclude that the value of unknown *P* remains within ϵ units from that of the reference value $P_{r}$.

It follows that the equivalence test includes two null hypotheses and a single alternative hypothesis, from which equivalence can only be claimed by rejecting both null hypotheses. The alternative hypothesis is defined as $P_{1}<P<P_{2}$ and the two null hypotheses should be defined as $0<P\leq P_{1}$ and $P_{2}\leq P<1$. Formally:$$\begin{array}{cc} H_{0}:\; & 0<P\leq P_{1}\;\mathrm{or}\;P_{2}\leq P<1\\ H_{1}:\; & P_{1}<P<P_{2},\;\mathrm{where}\;\left(0<P_{1}<P_{2}<1\right). \end{array}$$

Let us now review how the equivalence test formulated above can be used in practice to test experts’ calibration. First, we need to identify the intended coverage probability of elicited intervals $P_{r}$ and the accepted margin of deviation ϵ around the intended coverage probability, which defines the limits $P_{1}$ and $P_{2}$ of the alternative hypothesis $H_{1}$. The test statistic will be the number of successes out of a certain *n* number of Bernoulli trials, and in this context, it will be *x*, the number of calibration questions for which the expert’s credible interval covers the true value. This will then be evaluated relative to a rejection region (the calculations/evaluations necessary for obtaining the rejection regions are implemented in R, by Wellek, and shared as a Appendix A accompanying this paper) $\left(C_{1},C_{2}\right)$ that depends on the number of elicited intervals *n*.

When $x>C_{1}$ and $x<C_{2}$, then the null hypothesis is rejected (equivalence can be concluded), indicating that the expert is well-calibrated at the intended coverage probability $P_{r}$. When $x<C_{1}$ or $x>C_{2}$, the null hypotheses cannot be not rejected, indicating that the expert is not well-calibrated at the intended coverage probability $P_{r}$. On the boundaries of the rejection region (i.e., $x=C_{1}$ or $x=C_{2}$) the test is inconclusive and randomization is required to take a decision (the theoretical details of computing the rejection regions of the Wellek’s equivalence test are given in Appendix A).

Recall that, in the direct test, the expert is considered well-calibrated if $x=P_{r}\cdot n$. Let $C_{1}=P_{r}\cdot \left(n-1\right)$ and $C_{2}=P_{r}\cdot \left(n+1\right)$. Now the pattern of rejecting a hypothesis because the corresponding test statistic is within a region (in this case, $x>C_{1}=P_{r}\cdot \left(n-1\right)$ and $x<C_{2}=P_{r}\cdot \left(n+1\right)$) and not rejecting it otherwise is identical to the equivalence test (see, e.g., [13]) where the rejection region is collapsed to the single value $P_{r}\cdot n$. From this point of view, the direct test can be seen as an equivalence-style test for which the rejection region is determined independently of testing considerations such as size and power. Therefore, it is statistically meaningful to compare the performance of the direct test with that of the equivalence test when assessing experts’ calibration. To perform these comparisons, we need to assume that:experts have an underlying true level of calibration that is fixed for the given set of calibration questions, andexposure to calibration questions can be treated as independent and identically distributed experiments, so that the outcomes of the comparisons of the expert’s intervals against the known values are Bernoulli random variables.

The practicability of the above assumptions can be justified as follows. Generally, the calibration questions are prepared to assess the knowledge of experts in certain areas through a calibration experiment that conducts within a shorter period of time. Hence, the underlying true level of calibration of experts that depends on the knowledge in the corresponding areas of interest can reasonably be assumed as fixed at least for the duration of the calibration experiment.

It is also reasonable to assume that the calibration questions should be prepared to assess distinct knowledge in certain areas without having inter-dependencies between questions. Hence, the outcome of a given question can reasonably be assumed independent of the outcome of another question. Furthermore, following the first assumption, the true level of calibration of each question can be assumed fixed in a given calibration experiment. Therefore, the outcomes of the comparisons of experts’ intervals against the known values can be assumed to follow Bernoulli distributions as well.

To understand and compare the properties of the above described tests we will use simulated data (see Section 2.3). To further understand what these properties translate into or imply when it comes to real expert elicited data we will use data from two case studies (see Section 2.4).

### 2.3. Simulated Data

We consider a theoretical context where the true levels of coverage of expert elicited intervals are assumed equal to the intended coverage probabilities of elicited intervals. This allows us to carry out a power analysis of correctly identifying well-calibrated experts, both for the direct and equivalence tests. These power analyses will in turn provide insights into the most appropriate statistical technique to use when intending to reduce the error of identifying poorly calibrated experts as well-calibrated. We choose 80% and 90% intended coverage probabilities of elicited intervals for the analysis; these choices are inspired by the common practice in expert elicitation experiments (some of the following sub-sections are part of the Ph.D. dissertation of [15]).

The probabilities of correctly identifying experts as well calibrated from the direct and equivalence tests at 80% and 90% intended coverage probabilities depend on the number of elicited intervals (corresponding to the number of calibration questions). We varied this number to cover values from 10 to 250.

Consider simulating the data at a particular combination of a specified coverage probability (either 80% or 90%) and a specific number of elicited intervals for a given expert (e.g., 80% coverage probability and 50 elicited intervals). Each of the elicited intervals will have a random outcome which may or may not fall within the interval. This randomly generated outcome can be considered as a Bernoulli trial with a fixed probability of success that is equal to the assumed (in our example 80%) level of intended coverage of experts’ elicited intervals. It follows that the randomly generated outcomes of all the intervals (50 in our example) that include the realized values of quantities *x* (elicited as intervals from experts) can be considered as a binomial random variable with the same fixed probability of success (e.g., 80%) and the number of trials of the binomial distribution equal to the assumed total number of questions for which we elicit intervals (e.g., 50). Hence, random observations from binomial distributions can be used accordingly to generate total number of intervals that include realized values of quantities for any given expert in the analysis. It is important to note that even for a well-calibrated expert, the number of elicited intervals containing the realized values is still random (a realization of a binomial distribution).

Repeated simulations for a given expert allows us to compute the proportion of experts who are correctly identified as well-calibrated (at the specified combination of coverage probability and number of elicited intervals). Hence, the above discussed simulation procedure was repeated 25 times at each combination from Table 1. Here, the number of repeats, 25, can conceptually be considered as the number of experts in the analysis. However, it is important to carry out this analysis without considering a fixed number of experts. Therefore, the above process was repeated 100,000 times at each combination mentioned above to obtain the average proportions of experts who are correctly identified as well-calibrated from the direct and equivalence tests. These average proportions reasonably represent the probabilities of correctly identifying well-calibrated experts (the R program used for simulating and analyzing data can be found in a Appendix A).

The significance level α, and the acceptable margin of deviation ϵ around the reference value $P_{r}$ of the equivalence test were considered to be equal to 0.05. That is to say, we consider 0.05 to be an acceptable level of significance for the statistical analysis, and we consider a 0.05 deviation from the true level of coverage of elicited intervals to be reasonable enough to still consider an expert as well-calibrated.

The properties of the proposed tests will be presented and compared in Section 3.1. Before we turn to the presentation of results, we introduce two real-life data sets that will be used to draw practical conclusions and recommendations about these proposals.

### 2.4. Real-Life Data

The practical aspects of evaluating experts’ calibration are essential. Ideally, calibration tests should be as rigorous as possible from a statistical point of view, but unfortunately, the expert data and experimental design to collect it, are not always perfectly suitable for the most rigorous statistical analysis, due to practicalities.

As mentioned earlier, when expert elicitation experiments are conducted in practice, a limited number of calibration questions are asked to avoid extra elicitation burden for the experts. At least 10 calibration questions are usually recommended when eliciting quantities. The working theory is that even though calibration scores calculated based on so few variables will not be very reliable, at least they will pick up the major differences between experts and will be able to identify the very poorly calibrated ones. To illustrate the limitations of real data sets we chose two examples detailed in the next couple of sections. The two data sets are elicited using the different question formats (3-step and 4-step) discussed in Section 1.

#### 2.4.1. Four-Step Format Elicited Data

In this section, we very briefly introduce a data set described in detail in [16]. The data contains experts’ estimates of future abiotic and biotic events on the Great Barrier Reef, Australia. 58 experts have answered 13 calibration questions using the 4-step elicitation format. In this format, experts are asked for their confidence that the true value of the yet unknown quantity falls between the upper and the lower elicited bounds. This confidence is then used to define the credible interval. More often than not, the levels of assigned confidence for elicited credible intervals differ per question within a given set of expert’s answers. Hence, a transformation (usually extrapolation) is necessary to obtain credible intervals at a same level of confidence (please see [17] for details). Even though this transformation is necessary to allow the calculation of hit rate, it introduces another layer of noise into an already noisy process. The calculation of experts’ calibration, using the direct test, was undertaken in [16]. However, not all experts in this study answered all questions, and many experts varied the levels of confidence given per question.

We will use a subset of experts and questions to evaluate the proposed tests’ behavior in Section 3.2.1. The reason to further reduce an already small data set is to avoid further noisy signals influenced by extra assumptions needed when using transformations and accounting for a very small number of elicited intervals.

#### 2.4.2. Three-Step Format Elicited Data

The second data set used in this research was elicited for a Japanese geological disposal program. For the context and exact questions, we refer the reader to [18]. Suffice to say that 21 experts have answered 16 calibration questions using the 3-step elicitation format. It is important to note that no transformation of intervals is required in this analysis as all the intervals are elicited at a fixed level of 90% confidence from all the experts. In this elicitation however, the intervals are *assumed*
90% central credible intervals, with no guarantee that the experts’ estimates correspond to their internal 90% confidence. Since we are mostly interested in how well our proposed tests identify well-calibrated experts, we use only a subset of experts with the highest hit rates from this data set as well (see Section 3.2.2).

## 3. Results

### 3.1. Analysis of the Simulated Data

#### 3.1.1. Power Analysis

We first analyze the power of the direct and equivalence tests to correctly identify well-calibrated experts on eliciting 90% credible intervals. We calculate rejection regions $\left(C_{1},C_{2}\right)$ for the equivalence test, for different choices of *n* (see Appendix A for R code of the Wellek’s equivalence test). These are presented in Table 2.

Using these choices for *n* and the corresponding rejection regions we calculated and plotted (in Figure 1) the power of the direct and equivalence tests to correctly identify 90% well-calibrated experts at 90% true level of coverage of experts’ elicited intervals. The observed values of power of the direct test tend to decrease as the number of elicited intervals increases. This is a counter-intuitive result from a statistical point of view as the power of a test would ordinarily be expected to increase with increased number of samples. However, the probability of observing a single element decreases as the number of elements of a sample space increases. This is the reason behind decreased power when the number of elicited intervals is increased. On the other hand, the power of the equivalence test increases (as intuitively expected) with the increase in the number of elicited intervals. The power of the equivalence test is higher than the corresponding power of the direct test for 80 or more intervals.

#### 3.1.2. Different True Levels of Coverage of Intervals

We shall further investigate the properties of the direct and equivalence tests when testing experts’ calibration on eliciting 90% credible intervals when the true levels of coverage are different than 90%. This is interesting both from a theoretical point of view, as well as from a practical perspective, as it is often claimed (sometimes with proof offered) that experts are either over- or under-confident. Here, we assess the outcomes of the direct and equivalence tests when testing calibration, when eliciting 90% credible intervals at smaller true levels of coverage. This corresponds to the situation when experts’ assessments are overconfident. Figure 2 shows the probabilities of identifying the experts as 90% well-calibrated when the true levels of coverage are between 85% and 89%. The comparison between the tests is presented for different choices for the number of elicited intervals. Even though larger variations from true coverage may occur in practice, here we only investigate a 0.05 deviation from true coverage probabilities. The reason behind this choice is the intention to expose a potential problem of the equivalence test to produce higher probabilities of incorrectly identifying experts as well-calibrated compared to the direct test.

As discussed above, comparatively higher values of power to correctly identify 90% well-calibrated experts at 90% true level of coverage of elicited intervals can be obtained using the equivalence test compared to the direct test for 80 or more intervals. A trade-off seems to be that this is achieved at the expense of receiving comparatively higher probabilities of incorrectly identifying the experts as 90% well-calibrated at true levels of coverage remain between 85% and 89% from the test (see Figure 2). Furthermore, these probabilities increase with the increase of number of elicited intervals. The alternative hypothesis of the equivalence test $P_{1}<P<P_{2}$ to declare equivalence (that is, to declare an expert as well-calibrated) at 90% coverage probability with the acceptable margin of deviation ϵ=0.05 around the reference value P=0.9 contains values between $P_{1}=0.85$ to $P_{2}=0.95$ (refer the Section 2.2 for details). Therefore, from the equivalence test point of view, the probabilities of identifying experts as 90% well-calibrated at different true levels of coverage of elicited intervals within this range can be considered as the values of power (the probabilities of rejecting the null hypotheses when they are false) of the test at corresponding values in the rejection region of the test. Therefore, receiving comparatively higher probabilities of incorrectly identifying the experts as 90% well-calibrated at true levels of coverage remain between 85% and 89% here happens due to a characteristic of the equivalence test.

Let us now focus on the type I error probabilities (the probabilities of incorrectly accepting the alternative hypothesis when one of the null hypotheses is true) of the equivalence test given in Figure 2. Observe that type I error probabilities of incorrectly identifying the experts as 90% well calibrated are approximately equal to the size of the test 0.05 at 85% true level of coverage of intervals which is on the border of the rejection region of the test. Direct comparison of observed hit rates of experts’ elicited intervals with the intended coverage probability of credible intervals without considering the random variation of hit rates cannot be considered as testing experts’ calibration statistically.

However, we can implement the direct test as a special form of the equivalence test. By this we mean that in each case, the rejection region is bounded above and below, and we reject the null hypothesis if the observed value is within that region. In the case of the direct test, the region is a single value of P=0.9 in this instance.

If we consider the direct test as a special case of the equivalence test, then the probabilities given in Figure 2 of the direct test can be considered as type I error probabilities of incorrectly identifying the experts as 90% well-calibrated at the corresponding true levels of coverage of elicited intervals. Observe that the direct test has higher type I error probabilities of incorrectly identifying the experts as 90% well calibrated than the equivalence test for the considered true levels of coverage that are less than 90% at 50 intervals, even though the corresponding value of power to correctly identify well-calibrated experts was higher than the equivalence test as shown in Figure 1. It can also be observed that these type I error probabilities tend to decrease with an increase in the number of elicited intervals. However, this property of decreasing probabilities over the increase of number of elicited intervals is applicable to the probabilities of correctly identifying 90% well-calibrated experts at the 90% true level of coverage of elicited intervals as well. We can overcome this problem using the equivalence test if we can accept observing higher probabilities of incorrectly identifying the experts as 90% well-calibrated at true levels of coverage of elicited intervals are only within 0.05 deviation as observed above.

Even though the above problem can be overcome using the equivalence instead of the direct test, the equivalence test has the disadvantage of needing many more elicited intervals to obtain reasonably higher values of power to correctly identify well-calibrated experts. However, the application of the direct test implies that a reduced the number of elicited intervals is better (since this is how we obtain higher values of power to correctly identify well-calibrated experts). Rigorous statistical analysis advice should not encourage a reduced number of observations to obtain better performance of the statistical test, unless the additional observations do not add any valid information. This is however not the case, and hence, the results of direct test should be used cautiously.

Let us further review the outcomes of the direct test when a small number of intervals is elicited. Figure 3 indicates another problem of the direct test, and that is an increase in the type I error probabilities of incorrectly identifying the experts as 90% well-calibrated at the considered true levels of coverage of elicited intervals that are less than 90% with decreased number of elicited intervals. Observe that these probabilities are higher than the corresponding type I error probabilities at large number of elicited intervals shown in Figure 2 above.

Considering the direct test as a special form of the equivalence test enables the computation of the size of the test. Because the alternative hypothesis is defined using a single value, in this instance, the power and the size of the direct test are equivalent. The only value in the rejection region is P=0.9. Therefore, power can only be computed at P=0.9 as *Pr* (reject $H_{0}$|P=0.9). The alternative hypothesis with a single value of P=0.9 determines the two null hypotheses to be: $H_{0}:0<P\leq 0.9$ or 0.9≤P<1. The maximum type I error (or the size of the test) occurs at the border values of the null hypotheses. Here, the border value is P=0.9. Thus, the size of the test equals *Pr* (reject $H_{0}$|P=0.9). We consider the fact that the size is equal to the power (in the strict sense) as another reason to be suspicious of the direct test.

However, the equivalence test becomes a better option when a large (larger than 50) number of intervals is elicited. From a practical point of view, this equates with asking at least 50 calibration questions on top of the target questions in an expert elicitation. This is an incredible elicitation burden for the experts and, to our knowledge, it is rarely (if ever) undertaken. It is much more realistic for elicitations to contain between 10 and 30 calibration questions.

Figure 4 compares the size of the direct and equivalence tests at small number of elicited intervals. Observe that the size of the direct test decreases with the increase of number of elicited intervals. Therefore, if we apply the direct test to test experts’ calibration with small number of elicited intervals, the power of the test to correctly identify well-calibrated experts will not be sufficiently large. However, the size of the test which is equal to the value of the power should be considered as large from a statistical point of view. The equivalence test has lower values of power for the considered small number of elicited intervals with the fixed size of 0.05. If we increase the number of elicited intervals, the power of the equivalence test will be increased accordingly with the fixed size of 0.05. Therefore, the trade-off between the number of elicited intervals, power, and the size of the test is an important consideration in this context.

Similar patterns of results to the ones above were observed when testing experts’ calibration on eliciting 80% credible intervals of quantities. More importantly, the observed values of power to correctly identify well-calibrated experts on eliciting 80% credible intervals were lower than the corresponding values on eliciting 90% credible intervals for the considered range of number of elicited intervals for both the direct and the equivalence tests. This is an acceptable result in general as the binomial probability of obtaining 0.9 proportion of success under 0.9 success probability is higher than that of obtaining 0.8 proportion of success under 0.8 success probability for a given number of trials.

According to this property of the binomial distribution, it can also be shown that the power of the tests to correctly identify well-calibrated experts will further reduce if we reduce the intended coverage probabilities of elicited intervals more. Hence, the power of both the direct and the equivalence tests depends on the intended coverage probability of elicited intervals. This is an interesting and intriguing result.

#### 3.1.3. Improving on the Equivalence Test?

When discussing the implementation of the equivalence test using the number of intervals containing true values, we mentioned that the test is inconclusive for the boundaries of the rejection region. In these circumstances, randomization is required, and this randomization may be considered as another drawback. We therefore further analyze the situation when the non-randomized equivalence test with the rejection region of $C_{1}<x<C_{2}$ is used. Table 2 shows that the rejection regions for testing experts’ calibration on eliciting 90% credible intervals do not contain values greater than $C_{1}$ and less than $C_{2}$ for the number of elicited intervals less than or equal 80. Therefore, the non-randomized equivalence test with the rejection region of $C_{1}<x<C_{2}$ can only be applied for 100 or more intervals.

According to Figure 1, the equivalence test can only be considered more effective than the direct test for 100 or more intervals since the values of tests’ power are not considerably different at 80 intervals. Therefore, it seems meaningful to apply and observe the implications of the non-randomized equivalence test for 100 or more intervals.

Figure 5 plots the power of the direct and non-randomized equivalence tests. The tests have almost equal power at 100 intervals, with increasing power of the non-randomized equivalence test for more elicited intervals. However, its power is less than corresponding power of the equivalence test, due to the reduction of rejection regions.

Figure 6 indicates the probabilities of the direct and non-randomized equivalence tests to identify the experts as 90% well-calibrated when true levels of coverage of elicited intervals that are less than 90% for large number of elicited intervals. The type I error probabilities of incorrectly identifying the experts as 90% well-calibrated at 85% true level of coverage of elicited intervals of the non-randomized test are less than 0.05 for the considered number of elicited intervals. It implies that the significance level of the non-randomized test is less than the nominal value 0.05. Therefore, the test is conservative with reduced power of rejecting the null hypotheses when they are false.

From this perspective, the non-randomized equivalence test offers another (better) alternative to the direct test, when a very large number of intervals are elicited. This is a very appealing theoretical (rather than practical) alternative.

#### 3.1.4. Test Properties Established through the Simulation Study

The focus of this analysis was to assess the properties of several statistical tests that can be used to identify well calibrated experts.

The results of the above analyses show that the direct test (i.e., the direct comparison of experts’ hit rates) has substantial methodological problems. The test has low power to correctly identify well-calibrated experts and more importantly, the power decreases as the number of elicited intervals increases. This is a contradictory result from a statistical point of view.

The equivalence test of a single binomial proportion can be used instead to overcome these problems. However, power curves of the equivalence test show that many more elicited intervals are needed in this case, which is a practical impediment. Furthermore, the exact application of the binomial test for equivalence usually requires a randomized outcome if the observed coverage is on the border of the rejection region, an aspect of testing that may be distasteful to many analysts.

To summarize, testing whether experts are well calibrated or not proves to be a very challenging problem when balancing practicalities against statistical rigour. The direct test can be generalized to an equivalence test, which allows a formal test of the null hypothesis that the expert is *not* well calibrated. However, this requires a prohibitive number of calibration questions.

### 3.2. Analysis of the Real-Life Data

#### 3.2.1. The 4-Step Elicited Data

We reduced the scope of the present analysis and only examined experts who have answered a sufficiently large number of questions (out of the 13 questions) at either 80% and 90% levels of confidence, to avoid extra noise, artificially introduced through more transformations of the elicited data. The purpose of this analysis is to illustrates the implications of applying the direct and equivalence tests for a very small (but realistic) number of calibration questions.

First consider the analysis for the 80% credible intervals with the data given in Table 3. The experts ID’s are as presented in the original research.

Observe that the hit rates of these three experts are 69%, 75%, and 77%, respectively. No expert is well-calibrated under the direct test even though the last two experts are closer than the first one is to the intended calibration level of 80%.

When applying the equivalence test, the rejection region for the first expert is (10,11). Here, the number of intervals containing the true values is 9 which is outside the rejection region of the test. Hence, the equivalence test fails to reject the null hypothesis that the expert is not well-calibrated.

For the second expert, the rejection region is (9, 10) and the number of intervals containing the true values is 9. In this case, the test is inconclusive. Therefore, randomization is needed. After randomizing we cannot reject the null hypotheses (no equivalence) and can conclude that the expert is not well-calibrated. A similar analysis was required for the third expert, whose rejection region is (10, 11) and the number of intervals contacting the truth is 10. The randomization procedure led to the conclusion that expert four is not a well-calibrated expert either.

What is important to note here is that the equivalence test that considers the potential random variation of hit rates is unable to identify two experts with 75%, and 77% hit rates as well-calibrated at 80% intended calibration level probably due to the lack of elicited intervals leading to lack of power of the test.

Similar analyses were carried out for the subset of experts and questions corresponding to 90% credible intervals. In this case, a single expert (ID = 53a) answered 13 questions, with only 5 intervals containing true values. The corresponding expert was declared not well-calibrated from the direct test as the hit rate is 0.38%. The rejection region of the equivalence test is (11, 12) for this analysis. Hence, the conclusion from the equivalence test was the same. In this case, the two tests agreed and the result seems to be sensible as the observed hit rate is considerably different than the intended level of coverage.

#### 3.2.2. The 3-Step Elicited Data

For the present analysis we chose two experts with the highest hit rates as shown in Table 4. This is a practically sensible choice as if the experts with the highest hit rates cannot be declared well-calibrated, the other experts will also be declared as not well-calibrated.

The computed hit rates are 0.875 and 0.75 for the Exp16 and Exp21, respectively. Hence, both experts should be declared not well-calibrated at 90% confidence level, even though the decision is marginal for expert Exp16. The rejection region of the equivalence test with 16 questions at 90% confidence level is (14, 15). Therefore, Exp21 with 12 correctly elicited intervals covering the truth was declared not well-calibrated using the equivalence test. For expert Exp16, whose 14 intervals captured the truth, this value landed on the border of the rejection region of the equivalence test. Hence, the above discussed randomization procedure was applied. Based on that, the expert was declared not well-calibrated. These two experts were nevertheless the best calibrated experts from the expert group used in the elicitation detailed in [18]. The previous argument is also applicable to here. When the number of elicited intervals is fairly low, the equivalence test fails to conclude that experts are well-calibrated even though their observed hit rates are closer to the intended levels of coverage.

#### 3.2.3. Implications for the Real-Life Examples

The above analyses show that the application of the equivalence test could fail to produce significant results in realistic (hence limited) elicited data sets. This relates to the fact that the equivalence test has lower power to identify well-calibrated experts when the number of elicited intervals is low. Therefore, even though theoretically attractive, the proposed equivalence test needs to be tailored to more realistic situations. Before doing that however, a meta-analysis of all existing expert elicited data sets may shed more light on the properties of the proposed tests under sub-optimal conditions.

## 4. Discussion

The assessment of expert calibration cannot be done properly with a small number of elicited intervals. This idea, generally ignored in practice, is relevant to both the direct and the equivalence test. Furthermore, the direct test encounters theoretical difficulties that can be addressed using the equivalence test. Therefore, we can claim that the concept of applying the equivalence-based hypothesis testing procedure is more appropriate considering its statistical properties even though the need of a large number of elicited intervals to obtain acceptable levels of power is not feasible in practice.

We selected 0.05 for ϵ (the acceptable margin of deviation around the intended coverage probability of elicited credible intervals) of the equivalence test. We assumed that any less deviation from the true level of coverage is reasonable enough to consider a given expert as well-calibrated. The margin may be reduced or increased depending on a given context. The reduction of the margin will reduce the rejection regions of the test and reduce the power of the test to correctly identify well-calibrated experts.

It can be justified to consider 80% and 90% intended coverage levels as they are often used in practice. However, there is a limitation to consider only a few specific levels of confidence where the potential range of levels to be considered is decided by the acceptable margin of deviation around the reference value allowed in the test. It is not statistically sensible to increase the acceptable margin further as it leads to an increase in the type II error and a reduction in the power of the test. Overall, even when we consider a wider continuum, the overall conclusion of the analysis that the formal testing of the calibration of experts in an experimental setting may be prohibitively expensive would remain unchanged.

This does not however underrate the importance of the equivalence test in this context. It has already been shown that the equivalence test has the desired properties for large number of elicited intervals compared to the direct test. What is required further is the development of an equivalence style test that performs well for small number of elicited intervals. Hence, we suggest developing an equivalence style test and consider a wider continuum when assessing its properties as a future direction of this research.

## Figures and Tables

**Figure 1 entropy-24-00757-f001:**
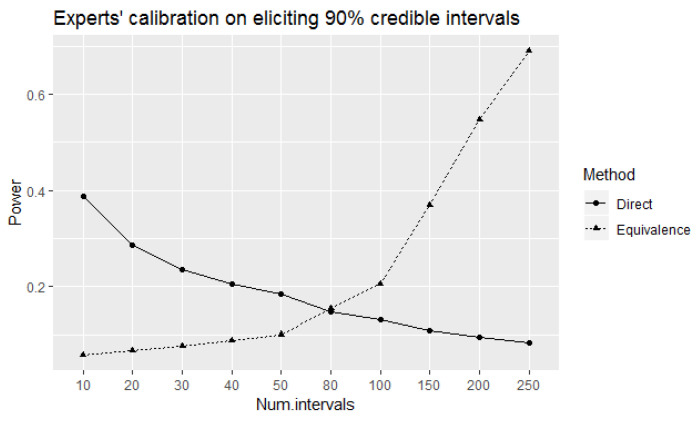
The power of the direct and equivalence tests to correctly identify 90% well-calibrated experts at 90% true level of coverage of elicited intervals.

**Figure 2 entropy-24-00757-f002:**
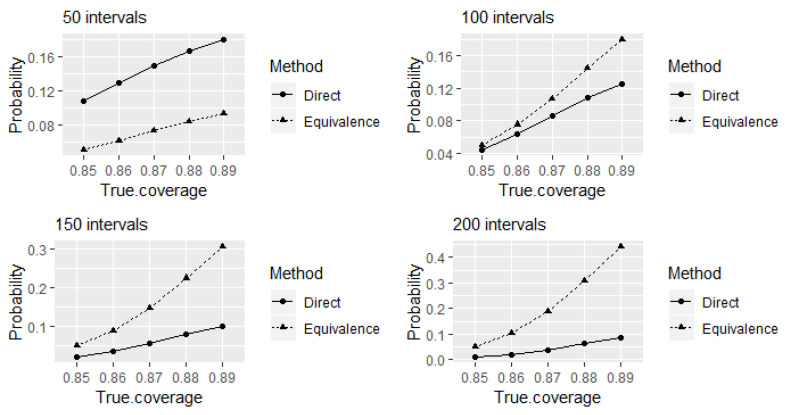
The probabilities of the direct and equivalence tests to identify the experts as 90% well-calibrated when true levels of coverage of elicited intervals that are less than 90%.

**Figure 3 entropy-24-00757-f003:**
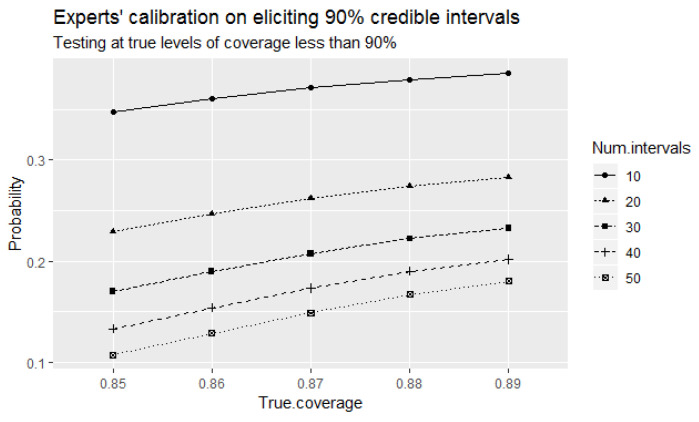
The probabilities of the direct test to identify the experts as 90% well-calibrated when true levels of coverage of elicited intervals that are less than 90% for small number of elicited intervals.

**Figure 4 entropy-24-00757-f004:**
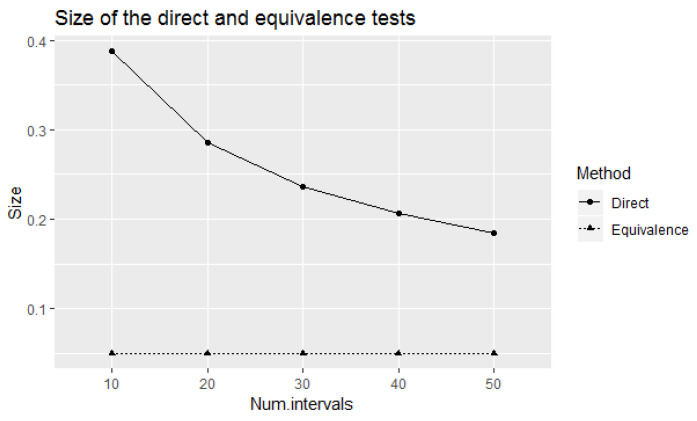
Size of the direct and equivalence tests in testing experts’ calibration on eliciting 90% credible intervals for small number of elicited intervals.

**Figure 5 entropy-24-00757-f005:**
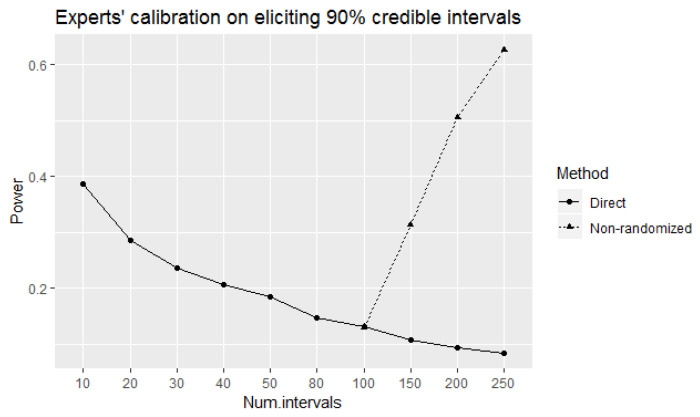
The power of the direct and non-randomized equivalence tests to correctly identify 90% well-calibrated experts at 90% true level of coverage of elicited intervals.

**Figure 6 entropy-24-00757-f006:**
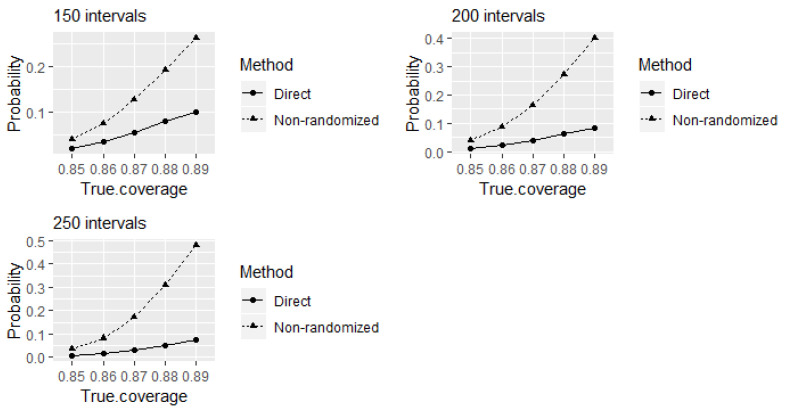
The probabilities of the direct and non-randomized equivalence tests to identify the experts as 90% well-calibrated when true levels of coverage of elicited intervals that are less than 90%.

**Table 1 entropy-24-00757-t001:** Coverage probability options and the number of elicited intervals used in simulating data.

Coverage Probability	Number of Elicited Intervals (*n*)									
80%	10	20	30	40	50	80	100	150	200	250
90%	10	20	30	40	50	80	100	150	200	250

**Table 2 entropy-24-00757-t002:** Rejection regions of the equivalence test.

Number of Elicited Intervals (*n*)	*C* _1_	*C* _2_
10	9	10
20	18	19
30	27	28
40	36	37
50	45	46
80	72	73
100	90	92
150	134	138
200	178	185
250	222	232

**Table 3 entropy-24-00757-t003:** Data from [16] used in the current analysis.

Expert ID	Num. Elicited Questions	Num. Intervals Covering the Truth
52b	13	9
54h	12	9
64i	13	10

**Table 4 entropy-24-00757-t004:** Data from [18] used in the current analysis.

Expert ID	Num. Elicited Questions	Num. Intervals Covering the Truth
Exp16	16	14
Exp21	16	12

## Data Availability

The full data sets that support the findings of this study are available on request from the corresponding author, due to ethics restrictions.

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
