# Peer review of "Are Experts Well-Calibrated? An Equivalence-Based Hypothesis Test"

_entropy, 2022, doi:10.3390/e24060757_

Round 1

Reviewer 1 Report

The authors consider the issue of calibration of experts in probabilistic assessments. This is a critical, and often overlooked, issue in real-world applications and any method in this area is of course of interest. I feel actually the author could more strongly make this point!!

The authors were quite apologetic about the actual contributions of the paper: saying that it was just an application of the method of Wellek, and I don't think they should! They are showing how a method could be used in a completely different setup and this is of course of interest, since it is also addressing such an important problem.

That said, I have major issues with the presentation of the material.

First, the paper feels disorganized and unstructured and I think authors should really review their use of sections/subsections etc. 

Second, the theoretical part should be described in much more details. I have some experience with expert judgments but I have never heard of these methods of Welleck. By reading the paper, I could still not understand them. What the methods proposed and reviewed actually do is kind of a mistery to me.

Third, the experiments seem to be rather comprehensive, but the lack of structure and a bit dispersive comments, made me lose track of what was happening. I believe the authors did a lot of good work, but the exposition did not make justice to it.

Last, the practical applications should be described in a bit more details. Although, they come from real-world applications they feel like toy examples.

So, overall I think there is some good material here, but the paper is way too far from being in a publication stage.

Reviewer 2 Report

I enjoyed reviewing this paper because in my field (medicine) expert judgements are often used by stringent methodology is rarely considered. The methodology seems to be fine, although as a researcher with just basic statistical background I cannot judge the soundness of the mathematical approach. 

I have just one major comment. The paper is written in a very technical way which may be difficult to understand for people outside this narrow field of research. The authors acknowledge that stating that "... that equivalence tests are not well known in the decision and forecasting literature." Unfortunately, I am afraid that this paper may go relatively unnoticed because researchers who would benefit from using expert opinion analysis will not be able to understand the implication. I wonder whether the authors could consider, perhaps in the supplementary information, presenting a step-by-step analysis that can be followed and reproduced by someone who is not an expert in mathematics. Even a more transparent analysis of the data in ref 17 could be helpful. 

Round 2

Reviewer 1 Report

Thanks for carrying out the review. The overall method is now much clearer. I appreciate the comprehensive discussion of both the merits and limitations of the proposed approach. 

The manuscript still presents various typos which will need to be fixed before publication.

At the bottom of page 4 you mention some assumptions of looking at the elicitation exercise as a Binomial experiment. How tenable are they? I would add an extra paragraph here discussing them.

On top of page 4 you introduce the epsilon defining the width of variation in the confidence level. Of course the results are affected by this choice. You give a discusssion about it in the conclusions, but I would like to see a more extensive one earlier on in the paper.

Reviewer 2 Report

No further comments, the revised version reads much better
